# Characterization of FCI (Fédération Cynologique Internationale) Grades for Hip Dysplasia in Five Dog Breeds

**DOI:** 10.3390/ani13132212

**Published:** 2023-07-05

**Authors:** Stefania Pinna, Aldo Vezzoni, Matteo Di Benedetto, Carlotta Lambertini, Chiara Tassani

**Affiliations:** 1Department of Veterinary Medical Sciences, University of Bologna, Via Tolara di Sopra 50, 40064 Bologna, Italy; matteo.dibenedetto3@unibo.it (M.D.B.); carlotta.lambertini2@unibo.it (C.L.); 2Clinica Veterinaria Vezzoni, Via delle Vigne 190, 26100 Cremona, Italy; aldo@vezzoni.it

**Keywords:** hip dysplasia, breed, radiographic parameters, dog

## Abstract

**Simple Summary:**

Hip dysplasia is one of the most frequently occurring orthopedic diseases in medium to large purebred dogs. For this reason, much attention is paid to screening programs for breeding animals. The Fédération Cynologique Internationale uses the same evaluation criteria regardless of breed. The aim of this study was to evaluate whether or not the evolution of a hip pathology in dogs with or without dysplasia was consistent with prior scientific knowledge in five breeds. In addition, whether there were significant radiographic differences between breeds which might require a breed-specific method for assessing the grade of hip dysplasia was investigated. Evaluations of 16 radiographic parameters analyzed using the Brass method were collected from 5 breeds: Labrador Retrievers, Golden Retrievers, German Shepherd dogs, Bernese Mountain dogs, and Rottweilers. No significant changes were found among the five breeds regarding the grade of hip dysplasia; however, some significant variations were found in the individual radiographic parameters suggesting that, although the criteria regarded all breeds, there were specific alterations which could be caused by the different morphologies, aptitudes and abilities of each breed.

**Abstract:**

The aim of this retrospective study was to verify whether the radiographic morphologic differences detected within the first three grades of hip dysplasia (A, B, C) of each of the five selected breeds and within the same breeds were statistically significant enough to require a breed-specific evaluation. A total of 422 technical evaluation forms of hip dysplasia (HD) in Labrador Retrievers, Golden Retrievers, Rottweilers, Bernese Mountain dogs, and German Shepherd dogs were obtained from the Fédération Cynologique Internationale (FCI) archive. The data were evaluated using a descriptive statistical analysis. In Labrador Retrievers, the craniolateral acetabular rim and femoral head position were already altered in unaffected dogs; however, within the various FCI grades, the most severe changes involved the conformation of the femoral neck. All the radiographic parameters of the Golden Retriever hips changed progressively and evenly. Significant radiographic changes between FCI grades were found in the German Shepherd dogs, and the alterations involving the acetabulum were more severe and appeared earlier than in the femoral head and neck. In the Bernese Mountain dogs, the most severe alterations were in the position of the femoral head and joint space while the femoral head and neck showed no significant progression between grades. All the radiographic parameters of non-dysplastic Rottweilers were normal; however, the progression of the primary signs was similar to the other breeds, although with lower severity. In conclusion, no significant prevalence of the radiographic features was observed for any specific breed. However, significant individual breed variations in the primary radiographic parameters were found between dogs with and without dysplasia which could be useful for better understanding the consequences of biomechanical differences between breeds.

## 1. Introduction

The Fédération Cynologique Internationale (FCI) is one of the World Canine Organizations which has recognized 356 breeds, each belonging to a specific country of origin. The scientific commission of the FCI and the country of origin of a breed have drawn up a detailed description of the ideal breed type or breed standard [1,2,3,4,5,6,7]. Moreover, the FCI has divided the breeds into 10 groups based on physical characteristics and aptitudes [8,9]. The FCI classifies the hip joint into five grades, from A to E, based on three different methods in order to highlight the main morphologic aspects of the hip joint: the Professor Wilhelm Brass, the Professor Mark Flückiger, and the Professor Malcolm B. Willis methods [10,11,12,13]. The Brass method is the one which is used for all breeds and consists of an analytic description of all aspects of the hip joint, defining each aspect as normal, with slightly altered or with pathological changes [5,10,14].

The morphologic and radiographic changes which occur during the development of HD were evaluated in a population of several FCI-recognized breeds and have recently been published [15]; however, the possible variations between breeds based on different morphologic characteristics were not investigated. The FCI evaluates all breeds using the same criteria for each grade of dysplasia without taking into account that breed-specific conformation may influence biomechanical differences and subsequent radiographic changes.

Radiographic study allows the identification of the primary and secondary changes in hip dysplasia. Alterations in the shape and depth of the acetabulum, the shape of the femoral head and neck, and the size of the Norberg angle are primary signs; osteophytic formations and exostosis are secondary signs. Both signs have been well described in previous reports [11,15,16,17,18,19]. The main aim of this study was to describe the changes in primary radiographic signs between the FCI grades of dysplasia: non-dysplastic (FCI-A), near-normal (FCI-B), and mildly dysplastic (FCI-C) hips of dogs belonging to five different breeds. The second aim was to discover whether there might be differences in radiographic changes among the most common breeds in order to eventually consider breed-specific criteria for assessment and scoring the HD in each breed.

## 2. Materials and Methods

A total of 422 technical evaluation forms of HD recorded between 2021 and 2022 were selected from the FCI archive of canine HD screening (Fondazione Salute Animale FSA, Cremona, Italy). The technical evaluation forms had been previously completed and graded for HD by at least two European College of Veterinary Surgeons (ECVS) and European College of Veterinary Diagnostic Imaging (ECVDI) diplomates. The selection was computer-randomized (simple random sampling) and, based on the study design, had to include 90 dogs of each of the following five breeds: Labrador Retriever (LR), Golden Retriever (GR), German Shepherd dog (GSd), Bernese Mountain dog (BMd), and Rottweiler (ROTT). The hips of the dogs were radiographed at a minimum age of 12 months for the LR, GR, and GSd breeds, and 15 months for the BMd and ROTT breeds, according to the FCI guidelines.

In addition, the inclusion criteria of the study design required that the 90 forms of each breed obtained from the archive, included 30 forms for each HD grade, FCI-A, FCI-B, and FCI-C, respectively, in order to obtain a homogeneous sample size. Sex, age, and weight were not restricted.

The FCI-D and FCI-E grades were excluded from this study since the presence of secondary signs of HD, such as osteophytes and severe deformities detectable in these grades of dysplasia could have invalidated the observation of the early evolution of the primary signs of HD which was one of the aims of this study.

The Brass method is a descriptive method which assesses sixteen radiographic parameters [15]. The evaluations of the radiographic parameters reported on the FCI forms, as well as the HD grade attributed to each dog, were reported in an Excel spreadsheet. All the data were processed and compared within each breed and between breeds (Appendix A).

The parameters reported on the technical evaluation form had previously been described [15] and were the following. Acetabular depth (AD), cranial acetabular margin (CAM), craniolateral acetabular rim (CLAR), and acetabular osteophytes (AOsts) were assessed in the acetabulum. The spherical femoral head (SFH), collar femoral head (CFH), and deformed femoral head (DFH) were evaluated in the femoral head (FH). For the position of the femoral head in the acetabulum, both the position of the femoral head (PFH) assessed as depth, slightly loose, and loose, and the femoral head center/dorsal acetabular rim (FHC/DAR) rated as center located medial, superimposed, and lateral to the DAR were considered. The femoral neck (FN) was assessed for several features: thin femoral neck (TFN), identifiable femoral neck (IFN) (only if normal), contours of the femoral neck (CFN), femoral neck osteophytes (FNOsts), and Morgan line in the femoral neck (MoFN) (only if present). The joint space (JS) was evaluated as concentric, slightly divergent, and divergent; finally, the Norberg angle (NA) was grouped as >105°, <105°, and <100°.

Each abovementioned radiographic parameter was arbitrarily rated as normal (N), slightly altered (SA), or as having pathological changes (PC) [15]. The AOsts, CFH, DFH, FNOst, and MoFN were assessed only if they were present (rated SA or rated PC) and were recorded as not applicable (na) in place of the N rating. On the contrary, an IFN was evaluated only if it was normal (N); therefore, it was recorded as not applicable (na) for the other ratings (SA or PC).

### Data Analysis

The data were analyzed using a statistical software program (MedCalc^®^ Software 16.8.4, Ostend, Belgium). All the data were tested for normality using the Kolmogorov–Smirnov test; however, the normality was rejected. A significance level of *p* < 0.05 was set. Appropriate summary and descriptive statistics (proportion, median, range, and 95% confidence interval [CI]) were calculated by breed and FCI grade of dysplasia for the following data: age, body weight, and sex. The Kruskal–Wallis test was used to compare age and weight between dogs of different breeds and, within each breed, between dogs assigned to different FCI grades. Post-hoc analysis was carried out using Dunn’s test. The multiple qualitative variables of the radiographic assessments were given in frequencies and percentages. The Chi-square test was used to compare the distribution of the categorical data of each breed. For each radiographic parameter, the associations between the FCI grades (A, B, C) were investigated, and the expected results were that the changes in severity level were the effects of the FCI grade. In addition, possible associations of the level of alteration of each parameter with breeds were investigated.

## 3. Results

### 3.1. Population

The Animal Health Foundation (FSA) archive from 2021 to 2022 included the FCI forms of 5439 dogs of several breeds. The FCI forms of 422 dogs were obtained from this archive. The study was designed to obtain the same number of dogs (n. 90) and the same number of FCI-A, FCI-B, and FCI-C grades (n. 30, respectively) for each breed; however, for two of the five breeds sorted from the database, namely, BMds and ROTTs, 30 subjects were not found for each required FCI grade. The dogs selected are listed in Table 1. The median ages and weights of the ROTTs and BMds were higher as compared with the other three breeds, and the differences were statistically significant (*p* < 0.001). The median weight of the LRs was lower when compared with all the other breeds (*p* < 0.001). The distribution of males and females did not differ significantly between the two groups (*p* = 0.262).

Of the 253 BMds recorded in the archive, only 21 were graded FCI-C and 9 dogs were not found in the archive. Of the 165 ROTTs, there were only 27 and 14 dogs in FCI-B and FCI-C, respectively; therefore, 3 dogs and 16 dogs, respectively, were not found in the archive. This deficit in numbers could be due to a limited demographic diffusion of these breeds.

No significant differences were observed for median age and weight within each breed group or for sex distribution among the three FCI grades, with the only exception being the GSds in which the median age was higher in the FCI-C group (15.5 (12–81) mo) than in the FCI-A and FCI-B groups (12.5 (12–50) mo and 12 (12–36) mo, respectively); the differences were statistically significant (*p* < 0.001).

The results obtained for each radiographic parameter were reported in frequencies and percentages for each breed and FCI grade. The percentages obtained are shown in Figure 1a–e.

All the data were reported as frequencies and percentages in Appendix A. The association between the change in rating and the FCI grade within each breed was investigated for each radiographic parameter. The most relevant results were the following.

### 3.2. Main Differences in the Evolution of the Radiographic Parameters among FCI Grades within Each Breed

For each radiographic parameter, any change between FCI-A, FCI-B, and FCI-C was verified using the Chi-square test. For greater precision, the expected results were considered; those changes between FCI grades in the same breed which were significant (*p* > 0.05) were not reported to avoid redundancy.

The results shown in Table 2 refer to radiographic parameters that did not change (*p* < 0.05) between two FCI grades. These were described and summarized in Table 2.

In the Labrador Retrievers, significant changes in the femoral neck radiographic parameters (the first three out of five parameters) were observed in the transition from normal (FCI-A) to near-normal (FCI-B) hips while they did not additionally worsen in dogs classified as dysplastic (FCI-C) (*p* > 0.05). In the Labrador Retrievers, the Norberg angle was normal (>105°) in the hips of the dogs classified as FCI-B (*p* = 0.48).

In the Golden Retrievers, the frequency of N and SA ratings of the thin femoral neck did not worsen from FCI-B to FCI-C (*p* = 0.21).

In the German Shepherd dogs, the Morgan line had already been observed in normal and near-normal hips; however, it did not progress significantly (FCI-A to FCI-B: *p* = 0.21).

In the Bernese Mountain dogs, various radiographic parameters of the acetabulum, and of the femoral head and neck showed changes not associated with the FCI grade, thus differing from the progression expected. The acetabular depth revealed *p* = 0.48 between FCI-A and FCI-B. The parameters, such as spherical femoral head, identifiable femoral neck, collar femoral head, and femoral neck osteophytes revealed *p* > 0.05 in the progression between FCI-B and FCI-C. The evolution of the Morgan line was not associated with FCI grade (*p* = 0.20 from FCI-B to FCI-C).

In the Rottweilers, the changes in the acetabular depth were slow from FCI-A to FCI-B (*p* = 0.43), the loss in the identifiability of the femoral neck between FCI-B and FCI-C was not significant. The Norberg angle, as in the Labrador Retrievers, was found to be normal (>105°) in FCI-B hips, and the reduction in the angle from FCI-A to FCI-B was not significant (*p* = 0.43).

In all the breeds, the parameters reporting the presence of osteophytes (such as acetabular osteophytes, collar femoral head, deformed femoral head, femoral neck osteophytes) only appeared in FCI-C; therefore, the comparative analysis with the previous FCI grade was not applicable. As an exception, an acetabular osteophyte rated SA was found in FCI-B in a Bernese Mountain dog; however, the number of hips affected was low and not statistically significant (*p* = 0.38) as compared with FCI-C.

### 3.3. Main Findings Recorded for Each FCI Grade in the Various Breeds

Each radiographic parameter was compared between breeds using a pairwise evaluation to check whether any differences in the same FCI grade could be an effect of breed. The list of significantly different parameters is included in Table 3.

Within each FCI grade (A, B, C), the main radiographic alterations which appeared to be associated with the breed and the most relevant percentage values and *p*-value are reported below (see Table 3 and Figure 1a–e).

#### 3.3.1. FCI-A

The craniolateral acetabular rim was mildly flattened (rated SA) by >20% in all breeds, except Rottweilers which were rated SA in only 5% (*p* < 0.05).

The femoral head center was superimposed (rated SA) for 21.7% of the Labrador Retrievers and was not statistically significant as compared to the German Shepherd dogs (*p* = 0.13); however, they differed from other breeds which had the center medial to the dorsal acetabular rim in >90% (rated N).

The joint space was slightly divergent by >40% in all the breeds except the Rottweilers which were rated SA in 11.7% (*p* < 0.05).

The other parameters were mainly normal in all the breeds.

#### 3.3.2. FCI-B

The acetabular depth became mildly flattened in >20% (rated SA) of the Labrador Retrievers and the German Shepherd dogs and in 13.3% of the Golden Retrievers; these values were significantly different from those of the Rottweilers and the Bernese Mountain dogs which kept their N rating in >95% of their hips.

The cranial acetabular margin changed from FCI-A to FCI-B with >20% rated SA (slight subchondral sclerosis) in all the breeds except the Labrador Retrievers and the Bernese Mountain dogs which had SA ratings of 10% and 20%, respectively, and a significance level of *p* < 0.05 as compared with the other breeds.

The craniolateral acetabular rim was mildly flattened by >70% in all the breeds except Rottweilers which were rated SA in 40% (*p* < 0.05).

The spherical femoral head was mildly small or flattened in >40% (rated SA) in all the breeds except Labrador Retrievers and Bernese Mountain dogs with *p* = 0.57 when compared with each other and *p* < 0.05 when compared with the other breeds.

The collar femoral head revealed a collar exostosis (rated SA) of <5% in the German Shepherd dogs, Bernese Mountain dogs, and Rottweilers. This parameter was not present in the previous FCI-A grade; therefore, statistical analysis was not carried out due to insufficient data.

The position of the femoral head was slightly loose (rated SA) in 88% of the Golden Retrievers, whereas, in the other breeds, it was present in 50–60%, and the levels of significance between the breed mentioned above and the other breeds were all *p* < 0.005.

The femoral head center was superimposed on the dorsal acetabular rim (rated SA) in >80% of the Labrador and Golden Retrievers (*p* = 0.79), whereas, in the other breeds, it was rated SA in <70% of the hips. The progression of this parameter found in Labrador and Golden Retrievers had *p* < 0.05 when compared with the Rottweilers and was found only in Labrador Retrievers when compared with the Bernese Mountain dogs (*p* = 0.038).

The Morgan line was present in a mild form in >20–30% of all the breeds, whereas it was present in only 8.3% of the German Shepherd dogs and had *p* = 0.07 only in the Rottweilers. In 3.3% of the Bernese Mountain dogs, the Morgan line was severe and was classified as PC.

The joint space was found to be slightly divergent in approximately 100% of all the breeds with no differences between them.

The Norberg angle was between 100° and 105° (rated SA) in >15% of the Golden Retrievers, German Shepherd dogs, and Bernese Mountain dogs. For the most part, the Labrador Retrievers and Rottweilers had a normal NA with *p* < 0.05 as compared with the other breeds.

Acetabular and femoral neck osteophytes were not found as early signs of osteoarthritis (OA) in any breed.

#### 3.3.3. FCI-C

The number of Bernese Mountain dogs and Rottweilers in FCI-C recruited from the archive was lower than that of other breeds: 21 dogs and 14 dogs, respectively.

The acetabular depth was mildly flattened (rated SA) in >50% of all the breeds, with the exception of Golden Retrievers and Bernese Mountain dogs, which showed a normal acetabular depth in 66.7% and 78.6%, respectively, with a significant distribution (*p* < 0.05) as compared with the other breeds.

The cranial acetabular margin was present with obvious subchondral sclerosis (rated PC) in 3.3% of the Golden Retrievers and >15% of the German Shepherd dogs and Rottweilers. They had *p* < 0.05 as compared with Labrador Retrievers and Bernese Mountain dogs, >40% of which remained rated N.

The craniolateral acetabular rim was severely flattened in >50% of the German Shepherd dogs; in the Rottweilers, it was severely flattened in only 10%. This was significantly different from the other breeds with the exception of Bernese Mountain dogs (*p* = 0.06).

Acetabular osteophytes were present in >15% in all breeds except Rottweilers and Bernese Mountain dogs (<10%). In the German Shepherd dogs, the osteophytes were present in 40% which had *p* < 0.05 as compared with the Labrador Retrievers and Bernese Mountain dogs.

The spherical femoral head remained normal (rated N) in >40% of the Bernese Mountain dogs, whereas, in the other breeds, it was <30%; they were evaluated with *p* < 0.05 as compared with the previous breed. All the breeds had an SA rating of >50%.

The collar femoral head was rated SA, i.e., some exostosis, especially >50% in the German Shepherd dogs. In the Rottweilers, there were fewer dogs with exostosis in the femoral head; however, 28.6% had obvious exostosis (rated PC).

A deformed femoral head was only found in 9.5% of the Bernese Mountain dogs.

The position of the femoral head remained deep (rated N) in 20% of the German Shepherd dogs and Rottweilers. It was slightly loose (rated SA) in almost 50% of all the breeds. It worsened to loose (rated PC) in almost 40% of the Labrador Retrievers, Golden Retrievers, and Bernese Mountain dogs. The overall evaluation of the three ratings indicated *p* < 0.05 for the latter three breeds as compared with German Shepherd dogs and Rottweilers.

The center of the femoral head was lateral to the dorsal acetabular rim (rated SA) in >50% of the Labrador and Golden Retrievers which were assessed using *p* < 0.05 as compared with the other breeds in which the SA rating was almost 30%.

The contours of the femoral neck were found to be predominantly rated SA (>70%) in all the breeds except Rottweilers in which the femoral neck contours had been completely lost (rated PC) in 20% of the hips. No significant associations were shown between the breeds.

Femoral neck osteophytes were present in all the breeds with an SA rating from 13.3 to 21.7%; in the Rottweilers, 20% were rated with severe osteophytes (rated PC) which was statistically significant (*p* < 0.05) as compared to the other breeds.

The Morgan line in the femoral neck was present, as the sum of the SA and PC ratings, in a smaller percentage in the Bernese Mountain dogs (30%) and a larger percentage in the Labrador Retrievers and Rottweilers (>70%). The severity of this radiographic parameter was found to be associated with Labrador Retrievers with *p* < 0.05 as compared with the other breeds, except the Rottweilers.

The joint space was slightly divergent (rated SA) in the German Shepherd dogs (80%), whereas, in the other breeds, it was divergent (rated PC) in >30% of the dogs. The German Shepherd dogs had *p* < 0.05 for the joint space as compared with the other breeds, except the Rottweilers.

The Norberg angle remained normal (rated N) in 60% of the Rottweilers. The other breeds showed a rather uniform distribution of the three ratings, with no significant prevalence.

## 4. Discussion

In the present study, a large number of radiographic parameters were collected and evaluated for each breed itself and among the five breeds. The main aim was confirmed since the variation of the individual parameters was associated with the FCI grade, with some exceptions. Some results were revealed to be non-significant for some radiographic parameters within the breed indicating that individual parameters remained constant at the transition of two of the FCI grades analyzed; therefore, changes were slow when compared to other parameters or other breeds. However, specific parameters progressed more slowly in relation to breed, and there were significant differences among them. In other words, the hypothesis that radiographic changes may be breed-related was confirmed for only some of the parameters, not for the overall assessment of HD grade.

The results showed interesting differences in the development of the primary radiographic signs between normal and dysplastic dogs among the various breeds.

### 4.1. Labrador Retrievers

In the non-dysplastic Labrador Retrievers (FCI-A), the relationship between the femoral head and the acetabulum was not perfectly normal as compared to the other breeds, with the craniolateral acetabular rim slightly altered, the center of the head superimposed on the dorsal acetabular rim, and the joint space slightly divergent (45% of the hips). The abovementioned did not influence the subsequent evolution to grade FCI-B, which was similar to that of the Golden Retrievers despite initially having the center medial to the dorsal acetabular rim. In Labrador Retrievers with near-normal hips (FCI-B), the progression was observed with the appearance of mild alterations (rated SA) involving all the parameters, with predominant worsening of the position of the head and conformation of the femoral neck. In FCI-B hips, the appearance of a slightly cylindrical shape (thin femoral neck: rated SA), associated with the partial loss of an identifiable femoral neck and sharp contours (contours of the femoral neck: rated SA), preceded the changes in the head which remained predominantly spherical in 66.7% of the hips. The subsequent progression of the femoral neck changed slowly between FCI-B and FCI-C and was not associated with grade (*p* > 0.05). The analysis of the Norberg angle was interesting as it remained normal (>105°) in FCI-A and FCI-B (*p* = 0.48) and showed a sudden change in hips with mild dysplasia (FCI-C). In summary, in the Labrador Retrievers, incongruence, although mild, was already evident in non-dysplastic dogs; however, the transition to dysplasia grade was evidenced by severe incongruence and morphologic changes in the neck rather than in the head.

### 4.2. Golden Retrievers

The non-dysplastic Golden Retrievers were similar to the other breeds. The position of the head in the acetabulum was already slightly loose in the majority of the dogs (88.3%) (position of the femoral head: *p* < 0.05) with near-normal hips (FCI-B); the depth of the acetabulum (acetabular depth: *p* < 0.05) remained normal in 66.7% of the dysplastic dogs (FCI-C). The Norberg angle showed similar changes when compared to the other breeds, with the exception of the Labrador Retriever. In summary, a gradual evolution of the radiographic parameters from non-dysplastic hips to dysplastic hips was observed in the Golden Retrievers in agreement with a previous study conducted on 316 dogs of different breeds among which 25.3% were Golden Retrievers [15].

### 4.3. German Shepherd Dogs

In the German Shepherd dogs, there was a significant progression among the FCI grades in all the parameters assessed. It was interesting to observe that, in the majority (80%) of dysplastic dogs (FCI-C), the joint space was only slightly divergent (rated SA), and only 18.3% had a markedly loose head position (position of the femoral head: rated PC). This was in contrast to the changes in the acetabular parameters and the appearance of acetabular osteophytes which were the most developed in this breed (40%). Therefore, the acetabular changes appeared earlier and were more severe than those of the head and neck. Early signs of OA and the greater severity of the alterations could be explained with a greater susceptibility to hip disease development [20] and, probably, to a less strict breed selection in years past [21]. The authors were unable to obtain evidence to confirm the presence of more severe acetabular osteophytes than in other breeds with the same FCI grade. However, in a recent study conducted on Labrador Retrievers and German Shepherd dogs [22], a different way of standing and a different trotting pattern of the two breeds was shown which could result in a different joint loading pattern. German Shepherd dogs perform more hip extension when trotting, suggesting that dogs with more extended hips might be more susceptible to HD or that HD results in a more extended hip [22]. The acetabular changes and osteophytes found in the German Shepherd dogs could be an expression of these biomechanical considerations.

### 4.4. Bernese Mountain Dogs

The Bernese Mountain dog was found to be statistically most similar to the Labrador Retriever, differing only in the sphericity of the head, the position of the head in the acetabulum, and the Morgan line which were only 3 out of the 16 parameters analyzed, assuming that the significant difference in weight and age at evaluation between the two breeds did not influence the development of dysplasia. In this breed, the acetabular depth remained normal in the majority (rated N: 78.6%) of the dysplastic dogs (FCI-C); however, this value did not appear to be associated with the evolution of the other acetabular parameters which were similar to the other breeds, although less severe, with the exception of the German Shepherd dog. On the other hand, it was interesting to observe that the Bernese Mountain dog, like the Labrador Retriever, showed the most severe alterations (rated PC) in the position of the femoral head and the joint space; however, the Labrador Retrievers had a predominantly altered acetabular depth (rated PC: 70%) in the dysplastic dogs. These discrepancies could indicate that the depth of the acetabulum was not related to the position of the head in the acetabulum but, presumably, the craniolateral acetabular rim which was found to be altered in all breeds except the Rottweiler would be responsible. From a biomechanical point of view, the craniolateral acetabular rim could be considered to be the cause and/or result of joint laxity in agreement with the literature which indicates joint laxity as the first manifestation of the evolution of HD [23,24,25]. The head and neck showed no significant progression from FCI-B to FCI-C, suggesting a probable lower predisposition to dysplasia in the Bernese Mountain dogs which could be supported by the lower number of dysplastic dogs found in the database or could be the cause of bias: there were only 21 FCI-C dogs out of 253 Bernese Mountain dogs recorded. This discrepancy could also be explained by the low demographic diffusion of this breed.

### 4.5. Rottweilers

The non-dysplastic Rottweilers all presented normal morphologic features, including the craniolateral acetabular rim and joint space which were altered in the other breeds. The progression of the primary radiographic signs of hip dysplasia followed the tendency of the other breeds but with lower severity. This could be explained according to the morphologic aspect of the breed. The presence of a large muscle mass typical of the breed standard classified by the FCI as Molossoid type could lead to a lower predisposition to dysplasia. The FCI defines the upper thigh as moderately long, broad, and strongly muscled [21]. It is no coincidence that only 14 of the 165 Rottweilers in the archive were classified as FCI-C grade. In 1995, Popovitch found a significantly higher risk of developing OA in the German Shepherd dog than in the Rottweiler, indicating a different susceptibility to HD between these two breeds [20].

In contrast to the Rottweiler, dogs of other breeds with normal hips (FCI-A) already showed a slight alteration in the cranial acetabular rim and joint space. These alterations could be the result of slight joint laxity in young dogs [26,27] since the dogs evaluated had a median age close to the minimum required by the FCI and, as they were medium-large breeds, muscular development reached maximum performance later than skeletal development [28]. However, in the Taroni study, it was reported that the clinical evaluation of hip laxity early in large breeds, such as Labrador Retrievers, Golden Retrievers, German Shepherd dogs, and large mixed breeds, was not correctly predictive of FCI grade at the minimum required age [29]. In Rottweilers, Vidoni et al. estimated 35 +/− 2 weeks to be the correct age for predicting HD development in Rottweilers [30,31].

In a previous study, the results of the radiographic alterations of 632 hip joints recorded on FCI forms were processed as in the Brass method [15]. The study was carried out on numerous breeds; however, they were only analyzed within each FCI grade from A to D. The earliest skeletal changes found in this study appeared in the acetabulum, but only in the craniolateral acetabular rim. Changes in the primary radiographic parameters evolved with predominant severity in the femoral head and neck; these were already evident in the FCI-B grade. The acetabulum was of normal depth in half of the joints in the FCI-C grade [15]. Greater specificity was given to the present study by evaluating five medium-large breed dogs commonly predisposed to HD and subject to official screening as part of a breeding selection program. Based on the investigation conducted, it was not possible to define a rigorous difference in the evolution of HD among the breeds or to evaluate a different predisposition based on the descriptive analysis of the results obtained.

Nevertheless, variations in individual radiographic parameters were revealed which we considered to be not enough to identify the radiographic features which could affect the overall assessment of the official FCI classification. In other words, assigning the grade of dysplasia on the basis of an individual parameter could lead to error.

Norberg angles less than 105° are consistent with hip laxity and represent a quantitatively expressible and rather objective parameter [32,33,34]. Owing to this very objectivity, there are numerous studies regarding the Norberg angle in both healthy and dysplastic animals [35,36], regarding the correlation between the Norberg angle and acetabular coverage [34,37], or regarding the femoral head center/dorsal acetabular rim ratio [33].

The results of the current study agreed with studies carried out on the value of the Norberg angle in different breeds in which, despite the physical similarity between Labrador and Golden Retrievers, the Labrador Retrievers had Norberg angle values much more similar to Rottweilers in both the FCI-A and the FCI-B grades while Golden Retrievers and German Shepherd dogs had a predominantly normal Norberg angle only in the FCI-A grade [36].

In a recent 2022 study involving Labrador Retrievers, Mostafa verified that 46 FCI-C dysplastic dogs had a mean Norberg angle value of =102° ± 1.5° [37], unlike the earlier study by Tomlinson in 2000 in which the Norberg angle was <100° [36]. This was similar to the results of the present study in which only 25% of the dysplastic Labradors had values <100° (rated PC) and 75% had values >100 (rated N + rated SA). This discrepancy could have been influenced by a slight difference in the number of dogs.

In 2010, Skurkova stated that the two main parameters used in the diagnosis of HD were the Norberg angle and the position of the center of the femoral head relative to the dorsal acetabular rim [33]. He verified a positive correlation and the concomitant presence of both these parameters with the same degree of dysplasia. In the current study, a correlation between parameters in the same individual was not carried out since the aim was to assess the main differences in the evolution of radiographic parameters between FCI grades within each breed and between breeds within the same FCI grade. However, analysis of the percentage values of the Norberg angle and femoral head center/dorsal acetabular rim in each breed in the FCI-B and FCI-C grades showed a difference between normal (rated N) and slightly altered (rated SA) values in favor of the NA, i.e., in disagreement with Skurkova’s study [33].

Some limitations were present in the study design. For example, the moderate and severe dysplasia grades, FCI-D and FCI-E, respectively, were excluded; however, this was deliberate since the aim of the study was to detect the early evolution of dysplasia; that is, the primary parameters between dogs with and without HD. An additional limitation was the need to process the categorical data; therefore, a high-power statistical analysis was not carried out. If the ratings were empirically changed from a descriptive scale to a numerical scale, the aim of the morphologic and descriptive assessment of the FCI grade could have been lost. Novel studies regarding the pairwise relationships of the radiographic parameters described by the Brass method used on the FCI form, in addition to those already described in the literature, will be useful in expanding knowledge of the evolution of HD.

## 5. Conclusions

In the present study, no statistically significant alterations were found in the primary radiographic aspects of the hip joints of the five breeds; therefore, the same criteria for assessing and establishing the degree of HD could be used for all breeds. However, the study conducted on individual radiographic parameters useful for the FCI in determining the degree of HD revealed some interesting differences which could be traced to the different morphologic shapes of the different breeds and thus to the possible different biomechanics manifested in standing and during the movement of each breed.

## Figures and Tables

**Figure 1 animals-13-02212-f001:**
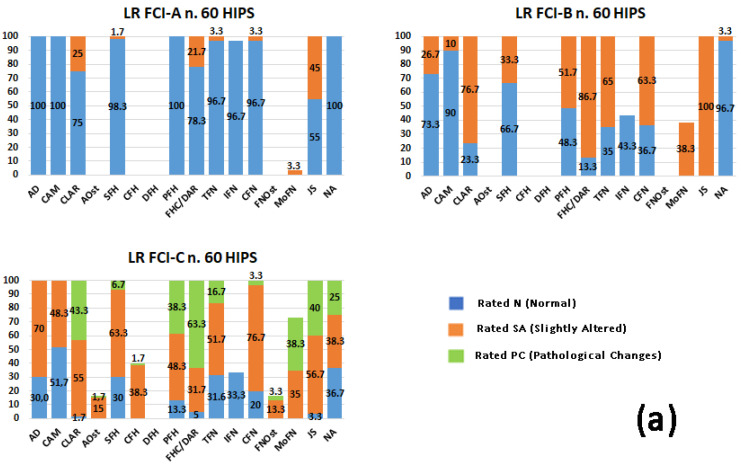
The graphs show the percentages (0–100 on the vertical Y axis) of the evaluation of the three severity levels (N, SA, PC) of the radiographic parameters in the FCI-A, FCI-B, and FCI-C grades of hip dysplasia according to Prof. Brass’s form. (**a**) Labrador Retrievers (LRs); (**b**) Golden Retrievers (GRs); (**c**) German Shepherd dogs (GSds); (**d**) Bernese Mountain dogs (BMds); and (**e**) Rottweilers (ROTTs). The blue column indicates the percentage of the normal rating (rated N), the orange column indicates the percentage of the slightly altered rating (rated SA), and the green column indicates the percentage of the pathological changes (rated PC). The acronyms of the 16 radiographic parameters on the horizontal X axis are described in Appendix B.

**Table 1 animals-13-02212-t001:** Data distribution of the five breeds regarding the number of hips for each FCI grade, and the sex, age, and weight of the dogs enrolled.

Breeds (n. Dogs)	n. Hips	FCI Grades n. Hips			Sex		Age (mo)			Weight (kg)		
		A	B	C	Male n. (%)	Female n. (%)	Median	Range (Min–Max)	95% CI	Median	Range (Min–Max)	95% CI
LR (90)	180	60	60	60	39 (43.3)	51 (56.7)	14 ^a^	12–68	13–15	29 ^a^	4–44	28–30
GR (90)	180	60	60	60	49 (54.4)	41 (45.6)	13 ^b^	12–96	12–16	31 ^b^	25–44	30–32
GSd (90)	180	60	60	60	45 (50.0)	45 (50.0)	13 ^b^	12–81	12–14	32 ^b^	23–50	30–33
BMd (81)	162	60	60	42	40 (49.4)	41 (50.6)	18 ^b^	15–58	17–20	42 ^c^	30–57	40–45
ROTT (71)	142	60	54	28	27 (38.0)	44 (62.0)	18 ^a^	15–62	17–19	42 ^c^	30–59	40–46

Legend. LR, Labrador Retriever; GR, Golden Retriever; GSd, German Shepherd dog; BMd, Bernese Mountain dog; ROTT, Rottweiler; A, FCI-A; B, FCI-B; C, FCI-C. ^a,b,c^: the different letters in the same column indicate statistically significant differences between groups (*p* < 0.001).

**Table 2 animals-13-02212-t002:** The list of radiographic parameters evaluated by the Chi-square test. Parameters which preserved their features as compared with the previous FCI grade were expressed by their *p*-value (*p* > 0.05). The parameters which had an expected change in the evolution of hip dysplasia at the transition between FCI-A, FCI-B, and FCI-C (*p* < 0.05) were marked by a dash “-”. The parameters which appeared only in the FCI-C grade were not analyzed (indicated by “na”).

Anatomical Locations	Radiographic Parameters	Breeds				
		LR	GR	GSd	BMd	ROTT
Acetabulum	AD	-	-	-	AB *p* = 0.48	AB *p* = 0.43
	CAM	-	-	-	-	-
	CLAR	-	-	-	-	-
	Aost	na	na	na	BC *p* = 0.38	na
Femoral head (FH)	SFH	-	-	-	BC *p* = 0.23	-
	CFH	na	na	na	BC *p* = 0.78	-
	DFH	na	na	na	na	na
Position of the femoral head in the acetabulum	PFH	-	-	-	-	-
	FHC/DAR	-	-	-	-	-
Femoral neck (FN)	TFN	BC *p* = 0.90	BC *p* = 0.21	-	-	-
	IFN	BC *p* = 0.35	-	-	BC *p* = 0.10	BC *p* = 0.21
	CFN	BC *p* = 0.06	-	-	BC *p* = 0.67	-
	FNOst	na	na	na	BC *p* = 0.20	na
	MoFN	-	-	AB *p* = 0.21	BC *p* = 0.20	-
Others	JS	-	-	-	-	-
	NA	AB *p* = 0.48	-	-	-	AB *p* = 0.43

Legend. LR, Labrador Retriever; GR, Golden Retriever; GSd, German Shepherd dog; BMd, Bernese Mountain dog; ROTT, Rottweiler; na, not applicable; A, FCI-A; B, FCI-B; C, FCI-C. The acronyms of the 16 radiographic parameters are described in Appendix B.

**Table 3 animals-13-02212-t003:** The distribution of the radiographic parameters which were statistically significant (*p* < 0.05) between breeds in the same FCI grade.

Breeds	GR	GSd	BMd	ROTT
LR	AD (C)	CAM (B, C)	AD (B, C) SFH (C) CFH (C) FHC/DAR (A, B) NA (B)	AD (B) CAM (B, C) CLAR (A, B, C) CFH (C) FHC/DAR (A, B, C) IFN (C)
	CAM (B, C)	CLAR (C)		
	SFH (B)	SFH (B)		
	PFH (B)	PFH (C)		
	FHC/DAR (A)	FHC/DAR (C)		
	TFN (C)	JS (C)		
	NA (B)	NA (B)		
GR		AD (C) CAM (C) CLAR (C) PFH (B) FHC/DAR (C) JC (C)	SFH (B, C) PFH (B)	CLAR (A, B, C) PFH (B, C) FHC/DAR (B, C) Aost (C) FNOst (C) JS (A) NA (B)
GSd			AD (B, C)	AD (B)
			CAM (C)	CLAR (A, B, C)
			CLAR (C)	CHF (C)
			SFH (B, C)	FHC/DAR (A)
			CFH (C)	TFN (B)
			PFH (C)	JS (A)
			JS (C)	NA (B)
BMd				AD (C) CAM (C) CLAR (A, B) SFH (C) PFH (C) JS (A) NA (B)

Legend. LR, Labrador Retriever; GR, Golden Retriever; GSd, German Shepherd dog; BMd, Bernese Mountain dog; ROTT, Rottweiler; A, FCI-A; B, FCI-B; C, FCI-C. The acronyms of the radiographic parameters are described in Appendix B.

## Data Availability

The data presented in this study are available in the Appendix A.

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
