# Peer review of "Characterization of FCI (Fédération Cynologique Internationale) Grades for Hip Dysplasia in Five Dog Breeds"

_animals, 2023, doi:10.3390/ani13132212_

Round 1
Reviewer 1 Report
Good study with interesting results
Title: add ‘hips’ and ‘dogs’ or ‘canine’. Characterisation would also be more appropriate than assessment.
Simple Summary:
There is no connection between the background and the aims of the study.
How can radiographic parameters result in biomechanical related parameters?
This sentence is not clear: ‘No breed prevalence was found in the assessment of the degree of dysplasia’ do the authors mean that breeds were not correlated to the degree of dysplasia?
Primary radiographic parameters should be mentioned before stating it as a result. This should be explained.
What is relevant? It should be significant or not, and clinically relevant or similar con be concluded.
Overall, this must be rephrased and should match the abstract.
Abstract:
The objective doesn’t match the one stated above: ‘to analyze the radiographic morphological differences…’ it seems to me like ‘biomechanic’ parameters should be removed.
Is the aim to analyse? Or correlate? Or describe? Probably the latter.
‘regarding the degrees’ is not clear again.
Replace: ‘with progression of the disease’ by between the grades’
Replace worst by severe or complex, etc.
‘A gradual evolution’ is not clear, what does it mean?
‘Significant progression between FCI grades’ in the next sentence is much clearer. Please rephrase the previous sentence.
The conclusion is unclear: ‘breed prevalence’ alone doesn’t mean much. Perhaps ‘a certain radiographic feature was not prevalent for any specific breed’ would be more accurate.
Remove the word trend. It is not a scientific not statistic term.
Mention the statistical tests
Overall, this section should be rephrased, the methods clearer, wording in the results checked and it should match the summary and the whole paper.
Conclusion should read similarly to the conclusion after discussion.
Intro:
This sentences are irrelevant: please remove. The Italian FCI member, the Italian National Kennel Club (ENCI: Ente Nazionale Cinofilia Italiana), collaborates with Fondazione Salute Animale (FSA), which is one of the official reading centers for hereditary breeding diseases, such as hip dysplasia (HD) [3-6].
The main purpose of this study was to detect the changes. First of all detect, assess, analyse or describe? This should be one thing across the whole manuscript.
If the Brass methods uses: normal, with slightly altered or with pathological changes, as it is done in the figures, please use the same categories in the aims, and not ‘near-normal’, which is not quite correct.
Define or cite primary radiographic signs in this section.
Results:
Line 190: what is it meant by ‘it did not evolve’? needs to be clarified or removed.
Discussion:
Line 373: please rephrase this sentence: ‘Neither from the analysis of the other parameters nor from a biomechanical point of view were the Authors able to extrapolate evidence to understand this difference as com- pared with other breeds.’ It is not well written.
Add this reference to this part of the discussion: https://pubmed.ncbi.nlm.nih.gov/33060670/
Conclusion is ok but the paragraph needs rephrasing.
The English Language of the simple summary and abstract should be improved. It is clearly less well written than the rest of the manuscript, where I only detected isolated mistakes regarding mainly readability.
Author Response
Dear reviewer, I have followed your suggestions, responding to each of your comments and editing the text. I hope my study now meets your expectations.

Reviewer 2 Report
I am sorry but I do not understand the point of this article. How does this information help us understand and diagnose hip dysplasia? How does this information help us identify dogs for breeding? How does this information help us prognose the future function of these dogs? Does the difference in changes have any relevance to information that we need. A tremendous amount of work went into this publication but I have to ask why? Please help me understand why this information is important?
Author Response
I hope that the modifications to the paper and the explanation can help you to better understand the main objective of this study and the interesting results that it produced.

Reviewer 3 Report
This is an interesting manuscript with potential novelty. the Fédération Cynologique Internationale (FCI) system is used in most European countries. The Brass method introduces the innovation of measuring the Position of the femoral head in the acetabulum. The results of the evaluation of 16 radiographic parameters analyzed using the Brass method were collected from five breeds. Interesting results are presented, as in the Labrador Retrievers, where incongruence, although mild, was already evident in non-dysplastic dogs. And the transition to dysplasia grade was evidenced by severe incongruence and morphological changes in the neck rather than in the head. Also, the earliest skeletal changes found in this study appeared in the acetabulum, but only in the craniolateral acetabular rim. However, some aspects should be improved. The description of the results and discussion is extensive and the authors must be careful not to contradict themselves. I have some suggestions and specific comments:
Simple Summary:
-May the authors explain the meaning of the sentence: ‘No breed prevalence was found in the assessment of the degree of dysplasia ‘? The same is in line 35 of the abstract. Maybe better to right:’ It was not possible to define a rigorous difference in the evolution of hip dysplasia among the breeds’.
Abstract
-Lines 25, 26: In the sentence’
In Labrador Retrievers, the craniolateral acetabular rim and femoral head position were already altered in normal dogs’ may the authors change ‘normal dogs’ to ‘unaffected dogs’?
Material and methods
-Lines 82,83: ‘The Brass method is a descriptive method which assesses sixteen radiographic parameters.’ Could the authors do reference to the article of Brass explaining method? Are there images to guide the evaluation of these 16 so specific parameters?
-Lines 83-85: ‘The evaluations of the radiographic parameters reported on the FCI forms' I have one question, was the evaluation of radiographs performed by different evaluators? How many? How expertized? This must be written in the document. There is some subjectivity in the evaluation, that must be assumed as a limitation.
-Line 99: Instead of ‘normal (N)’ could the authors use ‘without alterations’?
Data analysis
-Lines 108-110: ‘Appropriate summary and descriptive statistics (proportion, median, range, and 95% confidence interval [CI]) were calculated for the following data: breed, age, body weight, sex, FCI grade of dysplasia’.
Consider replacing by: ‘‘Appropriate summary and descriptive statistics (proportion, median, range, and 95% confidence interval [CI]) were calculated, by breed and FCI grade of dysplasia, for the following data: age, body weight, sex.’
Results
-I am a little bit concerned about the results presented in Table 2. What statistical test was used to obtain the results presented?
The title is constructed in the negative form and is confusing: ‘List of the radiographic parameters which did not reveal an association (p > 0.05) at the transition between FCI-A, FCI-B, and FCI-C. The statistically significant values (p < 0.05) were not shown as expected values (marked with dash "-"). The parameters which appeared only in grade FCI-C were not analyzed (reported as “na”).’
I think the authors must explain better the sentence: ‘The statistically significant values (p < 0.05) were not shown as expected values (marked with dash "-").‘ How did the authors define expected? The expected values were marked with dash (-)?
-Lines 205, 206: ‘An exception was the Bernese Mountain dog, which had an acetabular osteophyte SA rating in FCI-B but with an insignificant number.’ Explain better, considering the 5-grade FCI classification of canine hip dysplasia, there must be no signs of osteoarthrosis in FCI-B.
-Lines 252-253: I think this is a language problem. But could the authors explain better how 'The progression of this parameter appeared to be associated with Labrador and Golden Retrievers'. See other cases.
Discussion
-Lines 331,332: This is an interesting finding not surprising to me: ‘In the non-dysplastic Labrador Retrievers (FCI-A), the relationship between femoral head and the acetabulum was not perfectly normal’. Could the authors add to the sentence '…compared to other breeds', because this is actually the norm for that breed.
-Lines 355-359: Consider moving forward this sentence to the part of the discussion where the results of the breed BMD are discussed ‘The Bernese Mountain dog was found to be statistically most similar to the Labrador Retriever, differing only in the sphericity of the head, the position of the head in the acetabulum, and the Morgan line which were only 3 out of the 16 parameters analyzed, assuming that the significant difference in weight and age at evaluation between the two breeds did not influence the development of dysplasia.’
-I believe the information in Appendix A is important to understand the meaning of the terms used and so it must be published with the article.
Minor editing of English language required.
Author Response
Thank you for the suggestion. I have been careful to clarify the description of the M&M, Results and Discussion sections to improve readability. I have followed your suggestions, responding to each of your comments and editing the text. I hope my study now meets your expectations.

Reviewer 4 Report
In the paper entitled “Assessment of FCI (Fédération Cynologique Internationale) grades in five breeds”. The main purpose of this retrospective study was to analyze the radiographic morphological differences regarding the first three grades of hip dysplasia (A,B,C) in five selected breeds, which according to my knowledge is innovative because there are not many works that address this issue at the level of these radiographic changes in these specific breeds and in particular within the A, B and C animals. In this sense I consider this work original and relevant in this field. However as a veterinarian and clinician whenever I read a paper I question whether the information I have obtained will have practical use in the diagnosis and treatment of the animal in question and in the case of this work I am not sure if the results will effectively make a difference from a clinical point of view, so it would be good if the authors could clarify this point a little further. Nevertheless this paper addresses an important topic, especially given that canine hip dysplasia is considered to be the most common orthopedic condition diagnosed in the dog, in a different perspective than usual and all the work carried out to improve and refine the diagnosis of the disease is important.
With regard to the presentation of data collection, selection and analysis, possibly due to the large number of the data it becomes a little exhaustive and confusing, an aspect that could be improved.
With regard to the bibliography, this is an area in which there is a constant production of new material and most of the references are more than 5 years old so I think the bibliography could be improved by introducing more recent references
In resume the theme is relevant and as a global appreciation, I think that the work has good quality and relevance to be published after revision
Author Response

(The authors gave the same response as above.)

Reviewer 5 Report
animals-2381824
The paper is very interesting and covers a very interesting comparison of different dog breed dysplasia. The title should be more informative - please ad illness and specify your aim.
The main problem with this paper is connected with data selection and data analysis. Because of data pre-selection, the information (even randomized as stated in the methods) is strongly influenced in many fields. Especially if data were selected in the direction of investigated subject – number of FCI grades. The excluded D–E grades in % should also be at least mentioned in the paper. Probably that is the reason for different than expected statistical results – Authors describe individual cases, but then they write that it is not significant. It is not informative for the readers.
The authors mention in the limitation section that proper statistic is lacking, but they do not do anything with it. I can understand that some limitations may come from the equipment/environment that is not possible to be changed, but calculating new statistics is possible and the Authors should take it into account. The Chi-square is not enough for these data structures. Please check the papers on horse osteochondrosis for example when data are transformed in other scaling and analysis of variance with all necessary effects is used. As well as there are statistical programs with analysis of variance for data not normally distributed.
In Table 1 – it is visible that you were not able to find enough (according to your pre-selection) B and C grades in ROTT and BMd breeds and you write in the abstract that there are no differences in prevalence between breeds?
A major revision is needed. The paper is not acceptable in its present form. The descriptive part and discussion are very good parts. The aims are not clearly given from the beginning, material selection affects results, and methods are not adequate enough – so the conclusions are confusing at the end.
Please make your paper clear and correct as much as possible through the text.
Detailed remarks:
L 50- the method of evaluation is not significant for research repeatability - not names, please exclude from the introduction, if you like you can use acknowledgments
L 62 – it should be classified and described in a detailed manner
L 64 – do not write on the musculoskeletal conformation –as you did not check it in the paper, just breed differences would be fine. Differences you may have could be connected not necessarily from the muscle and skeleton structure.
L 72 – the age effect should be taken into account in the analysis of variance (probably more data could be taken into account at least for the basic grade evaluation analysis)
L 76 – as above all these effects have to be taken into evaluation as they are of great importance for skeleton disorders, overwise the differences observed by simple chi-square are biased – and you may not discuss a single effect
L 84 -86 but not with the correction for very significant effects
L 86 – it would be useful to give the supplement with your parameters/locations
L 99- description of these scaling is needed –at least a citation
L 114 – not clear; effects? – no associations
L116 – 117 – effect of something on something – not association
L 120 – such preselection affects results – correct statistical analysis is necessary; first on all data and with all grades only; then descriptive data on all if possible or selected with necessary but with proper statistics
L 324 – interesting objective, however not given earlier; is different like the one stated in the introduction - please write more in the introduction and put more info on it in a new version of the paper – such an aim does not need analyses of variance (but analysis of associations –correlations) but all paper should be described another way.
It seems that you have two different aims – first if there are differences between breeds in FCI grade, and second, if the grades are evaluated the same way between breeds. You have to be clear in it throughout all your paper as both aims need different statistics and comparisons.
Author Response

(The authors gave the same response as above.)

Round 2
Reviewer 2 Report
Thank you for your work making the changes that I recommended